# *t*-BuOOH/TiO₂ Photocatalytic System as a Convenient Peroxyl Radical Source at Room Temperature under Visible Light and Its Application for the CH-Peroxidation of Barbituric Acids

Elena R. Lopat'eva ⬤, Igor B. Krylov *⬤ and Alexander O. Terent'ev *⬤

N. D. Zelinsky Institute of Organic Chemistry, Russian Academy of Sciences, 47 Leninsky Prospekt, Moscow 119991, Russia; elena.lopatyeva@gmail.com
* Correspondence: krylovigor@yandex.ru (I.B.K.); alterex@yandex.ru (A.O.T.)

**Abstract:** TiO₂ is one of the most promising heterogeneous photoredox catalysts employed in oxidative pollutant destruction, CO₂ reduction, water splitting, disinfection, solar cell design and organic synthesis. Due to the wide bandgap of TiO₂, visible light energy is not sufficient for its activation, and electron/hole pairs generated upon UV irradiation demonstrate limited selectivity for application in organic synthesis. Thus, the development of TiO₂-based catalytic systems activated by visible light is highly attractive. In the present work we demonstrate the generation of *t*-BuOO• radicals from *tert*-butylhydroperoxide catalyzed using commercially available unmodified TiO₂ under visible light. This finding was used for the highly selective CH-peroxidation of barbituric acids, which contrasts with the behavior of the known TiO₂/H₂O₂/UV photocatalytic system used for deep oxidation of organic pollutants.

**Keywords:** titanium dioxide; heterogeneous photocatalysis; visible light photocatalysis; photoredox catalysis; *tert*-butyl hydroperoxide; barbituric acids





## 1. Introduction

Heterogeneous photocatalysis has recently gained a lot of attention as an ideal tool for green chemistry [1]. It utilizes the energy of light using non-toxic environmentally friendly photocatalysts that can be easily recycled and reused. Titanium dioxide (TiO₂) is the most widely used heterogeneous photocatalyst due to its excellent chemical stability, low toxicity, moderate cost and availability [2]. The most profound applications of TiO₂ are currently solar cell design [3,4], the oxidative destruction of pollutants [1,5–7], photodisinfection [8], hydrogen generation [9–12], CO₂ reduction [2,13] and water splitting [14]. One of the most important challenges for TiO₂ application in organic synthesis is to shift its activity form UV to visible light [15–18]. Visible light is an attractive energy source for chemical transformations, as it is responsible for the greatest part of sunlight's irradiation power compared with UV light. UV-mediated photochemistry is frequently associated with additional safety precautions, expensive light sources (compared with visible light) and the need for UV-transparent quartz glassware. Moreover, most organic compounds, including solvents, absorb UV radiation, which leads to lower energy efficiency and unwanted photochemical side reactions. Visible light quanta have lower energy and, thus, are more promising for the development of selective and widely applicable photochemical synthetic methods based on suitable photocatalysts. Numerous attempts were directed to the photosensitization of titanium dioxide in the visible light region by another semiconductors [19–22], organic photocatalysts [23] or via element doping [24–29]. Another opportunity is to use organic compounds, especially bearing hydroxyl or carboxyl groups, such as phenols [30,31], salicylic acid [32,33], carboxylic acids [34–36] or N-hydroxyphthalimide [37,38], which can modify the TiO₂ surface to absorb visible light. In our work, we discovered that peroxides can also lead to visible light sensitization: the mixing of TiO₂ suspension in MeCN

with *tert*-butylhydroperoxide resulted in a change in color from white to pale yellow (the appearance of the mixture can be found in the Supplementary Materials).

In our previous work, it was found that some hydroperoxides, in particular ethylbenzene hydroperoxide [37] and *tert*-butyl hydroperoxide (Scheme 1a) [38], can be decomposed on the TiO$_2$ surface when irradiated with visible light (443 nm). Previously, studies of peroxide-TiO$_2$ photochemistry were mainly focused on H$_2$O$_2$ and UV light usage [39–45]. Data on photoreactions involving unmodified TiO$_2$ and organic peroxides [46], especially under visible light irradiation, are rare (Scheme 1a) [25,47].

**Scheme 1.** Organic peroxides in TiO$_2$-photocatalyzed oxidative transformations. Previous works on benzylic alcohol oxidation [47], oxidative coupling of alcohols with *N*-hydroxyphthalimide [25], and cross-dehydrogenative coupling of ethers with electron-deficient heteroarenes [38] (**a**) and present work (**b**).

Organic peroxides are widely used as green metal-free oxidants [48–50] and O-reagents for cross-dehydrogenative C–O coupling reactions [51–58]. As a rule, the generation of peroxyl radicals from organic peroxides requires high temperatures [48,49,56,57] or salts of transition metals, such as Mn [55], Fe [51,58,59], Co [52], Ni [60], Cu [50,53,54,59] or Ru [61]. Direct photolysis of organic peroxides requires UV [62–65] or high-power white light [66], so various organic photoredox catalysts such as Eosin Y [67,68], Rose Bengal [69], Rhodamine B [70] or metal complex dyes, including Ir and Ru complexes [71,72], were proposed for the visible-light-induced photodecomposition of peroxides. However, examples of heterogeneous photocatalysts used to generate peroxyl radicals are still rare [38,47,73–75].

In this work, we investigated the decomposition of different organic peroxides using commercially available unmodified TiO$_2$ under visible light and demonstrated that the *t*-BuOOH/TiO$_2$ system can be used for the generation of peroxyl radicals under visible light (Scheme 1b).

Medicinally relevant compounds, barbituric acids, were chosen as substrates. Barbituric acids represent an important class of biologically active compounds [76], the direct functionalization of which is currently a hot topic for research [77–79]. The proposed methods for the peroxidation of barbituric acids often deal with transition metal catalysis [53,80] or proceed at high temperatures [53], and therefore the development of mild metal-free conditions for the peroxidation of barbituric acids is an attractive task.

## 2. Results

### 2.1. Study of Organic Peroxide Decomposition on TiO$_2$ under Visible Light

In the first step, we studied the decomposition of organic peroxides of various classes (hydroperoxides, dialkyl peroxides, diacyl peroxides and peroxyacids) under visible light in a suspension of commercially available nanosized titanium dioxide (Table 1). Based on our previous research [37,38], we used TiO$_2$ Hombikat UV100 with a high specific surface area (300 m$^2 \cdot$g$^{-1}$) as a heterogeneous photocatalyst and a 443 nm blue LED with 10 W input power as a light source. In all cases, corresponding alcohols **2** (or benzoic acid, in the case of mCPBA) were detected as the main decomposition products.

**Table 1.** Photodecomposition of organic peroxides on TiO$_2$ under blue light irradiation.

RO—OR or RO—OH **1** (1 mmol) → TiO$_2$ (10 mg), 10 W Blue LED, MeCN (2 mL), 25 °C, 3 h → ROH **2**

| Entry | Peroxide | Conversion, % | Yield [a] 2, % |
|---|---|---|---|
| 1 | *t*-BuOOH | 81 | 58 |
| 2 [b] | *t*-BuOOH | 0 | nd |
| 3 [c] | PhC(OOH)(CH$_3$)$_2$ (80%) | 63 | 31 |
| 4 | BzOOBz (75%) | <5 | trace |
| 5 | mCPBA (75%) | 40 | 40 |
| 6 | *t*-BuOO*t*-Bu | <5 | nd |

[a] Yields were determined with $^1$H NMR using C$_2$H$_2$Cl$_4$ as internal standard, mixture composition was additionally confirmed with $^{13}$C NMR. [b] Dark conditions. "nd" stands for not detected by $^1$H NMR. [c] 11% of acetophenone was formed.

Hydroperoxides (*t*-BuOOH and cumyl hydroperoxide) were found to be the most susceptible to photodecomposition (entries 1, 3). *Tert*-butyl hydroperoxide (TBHP) decomposes at a higher conversion than cumyl hydroperoxide (Table 1, entries 1, 3). This fact can be attributed to the higher lipophilicity of cumyl hydroperoxide compared with TBHP and, thus, the lower degree of adsorption of cumyl hydroperoxide on the TiO$_2$ surface. It should be noted that TBHP did not decompose in the absence of light (entry 2). Another important note is that the fraction of the beta-decay product, acetone, for the *tert*-butoxyl radical is negligible (no acetone signals are observed in the $^1$H and $^{13}$C NMR spectrum), while for cumyl hydroperoxide, about a quarter of the alkoxyl radicals undergo β-scission with the formation of acetophenone and the release of the methyl radical. Dibenzoyl peroxide (BzOOBz, entry 4) is rather stable under reaction conditions. The traces of benzoic acid observed in the reaction mixture were present in the starting commercial BzOOBz. Meta-chloroperbenzoic acid (mCPBA, entry 5) decomposes mainly to meta-chlorobenzoic acid (mCBA). For the correct determination of mCPBA conversion and the yield of meta-chlorobenzoic acid, $^1$H NMR spectra were recorded both from the starting mCPBA and reaction mixture in MeCN with the internal standard C$_2$H$_2$Cl$_4$. The initial mCPBA is of 75% purity (contains mCBA and water), and the molar ratio mCPBA/mCBA is 5.1/1, while after the reaction the molar ratio mCPBA/mCBA decreases to 1/1, which means that conversion of mCPBA is 40%. Di-*tert*-butyl peroxide turned out to be stable under the experimental conditions (entry 6), which gives hope that other dialkyl peroxides, in particular target peroxidation products, will not undergo decomposition on TiO$_2$ under visible light.

Hydroperoxide decomposition on TiO$_2$ can start from either single-electron oxidation (Scheme 2, **A**) leading to peroxyl radical, or single-electron reduction, (Scheme 2, **B**) leading to alkoxyl radical. Peroxyl radicals can undergo bimolecular decay (Scheme 2, **C**) with the formation of alkoxyl radicals [81–84]. In turn, alkoxyl radicals can abstract hydrogen atoms from the comparatively weak O–H bond of hydroperoxides with the formation of alcohols and peroxyl radicals (Scheme 2, **D**) [82]. The formation of ketone products (observed for

cumyl hydroperoxide, Table 1, entry 3) is a sign of the β-scission process in the alkoxyl radical (Scheme 2, **E**) [83].

**Scheme 2.** Plausible schematic mechanism of $TiO_2$-photocatalyzed decay of organic hydroperoxides.

To prove that *tert*-butylperoxyl radicals are formed in $TBHP/TiO_2$ systems under visible light, we set up radical trapping experiments with 1,1-diphenylethylene **3** and 2,6-di-*tert*-butyl-4-methylphenol (BHT, **6**) (Scheme 3). The isolation of *tert*-butylperoxy products **4** and **7** unambiguously confirmed the presence of *tert*-butylperoxyl radicals in the system.

**Scheme 3.** *Tert*-butylperoxyl radical trapping (isolated yields are given. NMR yields are given in parenthesis).

Summing up all the data obtained, *tert*-butyl hydroperoxide is the most suitable source of peroxyl radicals and can be used for the peroxidation of substances containing relatively weak CH bonds with the formation of unsymmetrical organic peroxides that are hoped to be stable under the reaction conditions.

### 2.2. Synthesis of Alkyl Peroxides from Barbituric Acids

In the next stage, the peroxidation of barbituric acids using a *tert*-butyl hydroperoxide/$TiO_2$ system under visible light was studied using 1,3-dimethyl-5-benzylbarbituric acid **8a** as a model substrate (Table 2).

**Table 2.** Optimization of amount of *t*-BuOOH, solvent and time for the peroxidation of **8a**.

| № | Changes to the General Conditions | Conversion, % | Yield [a] 9a, % |
|---|---|---|---|
| 1 | none | 100 | 54 |
| 2 | no light | 0 | nd |
| 3 | no TiO$_2$ | 0 | nd |
| 4 | air atmosphere | 100 | 37 |
| 5 | reaction time 1 h | 79 | 38 |
| 6 | reaction time 2 h | 94 | 40 |
| 7 | reaction time 8 h | 100 | 55 |
| 8 | 1 equiv. of *t*-BuOOH | 70 | 24 |
| 9 | 2 equiv. of *t*-BuOOH | 81 | 35 |
| 10 | 6 equiv. of *t*-BuOOH | 100 | 50 |
| 11 | 5 mg of TiO$_2$ | 84 | 31 |
| 12 | 20 mg of TiO$_2$ | 100 | 46 |
| 13 | DMSO as a solvent | 59 | 7 |
| 14 | DMF as a solvent | 100 | 14 |
| 15 | AcOH as a solvent | 100 | 32 |
| 16 | DCE as a solvent | 78 | 29 |
| 17 | g-C$_3$N$_4$ instead of TiO$_2$ | 56 | 23 |
| 18 | scaled up to 1 mmol of **8a** | 100 | 56 |

[a] Isolated yields are given. "nd" stands for not detected by [1]H NMR.

Under general conditions, we obtained the desired product **9a** in a 52% yield (entry 1). In the absence of light or photocatalyst, no conversion of **8a** was observed (entries 2, 3), proving that the target reaction is indeed a photocatalytic process. Carrying out the reaction in an air atmosphere led to a decrease in **9a** yield (entry 4), presumably due to the interaction of C-centered radicals formed from **8a** with molecular oxygen. Reducing the reaction time to 1 or 2 h (entries 5 and 6) led to the incomplete conversion of **8a**. A longer reaction time than required for complete conversion (8 h, entry 7) did not lead to a decrease in the yield of **9a**, which indicates that the product **9a** is stable under the reaction conditions. The use of a two-fold excess or equimolar amount of *t*-BuOOH resulted in the incomplete conversion of **8a** and low yields of **9a** (entries 8 and 9). Increasing the excess of *t*-BuOOH over four equivalents did not lead to a further increase in the yield of **9a** (entry 10). The decrease in the TiO$_2$ loading had a negative effect on the yield of the target product **9a** (entry 11). The increase in TiO$_2$ dosage from 10 mg (standard conditions) to 20 mg (entry 12) led to slight decrease in **9a** yield. Switching the solvent from MeCN to other polar solvents (DMSO, DMF, AcOH) led to lower yields of the target product (entries 13–15). 1,2-Dichloroethane (DCE) was less effective due to its immiscibility with water from aqueous *t*-BuOOH, leading to the aggregation of TiO$_2$ particles in the water phase (entry 16). Another widely used visible-light active photocatalyst g-C$_3$N$_4$ can also promote the photochemical peroxidation of barbituric acid **8a**, but much less efficiently than TiO$_2$ (entry 17). At last, we studied the scalability of the developed procedure (entry 18). It turns out that scaling up to 1 mmol of **8a** led to a slight increase in **9a** yield; therefore, the conditions of entry 18 were chosen as optimal for the peroxidation of other barbituric acids. As one can note, the yields of peroxide **9a** in many cases were significantly lower than conversions of the starting barbituric acid **8a**. The main side product observed with NMR analysis of the reaction mixture was 5-hydroxylated barbituric acid **10a** (5-benzyl-5-hydroxy-1,3-dimethylpyrimidine-2,4,6(1*H*,3*H*,5*H*)-trione). The isolated yields of the side product **10a** are not reported because it was not isolated in an analytically pure form in all cases.

In the next step, we extend our method to the synthesis of other 5-*tert*-butylperoxybarbituric acids **9** (Scheme 4).

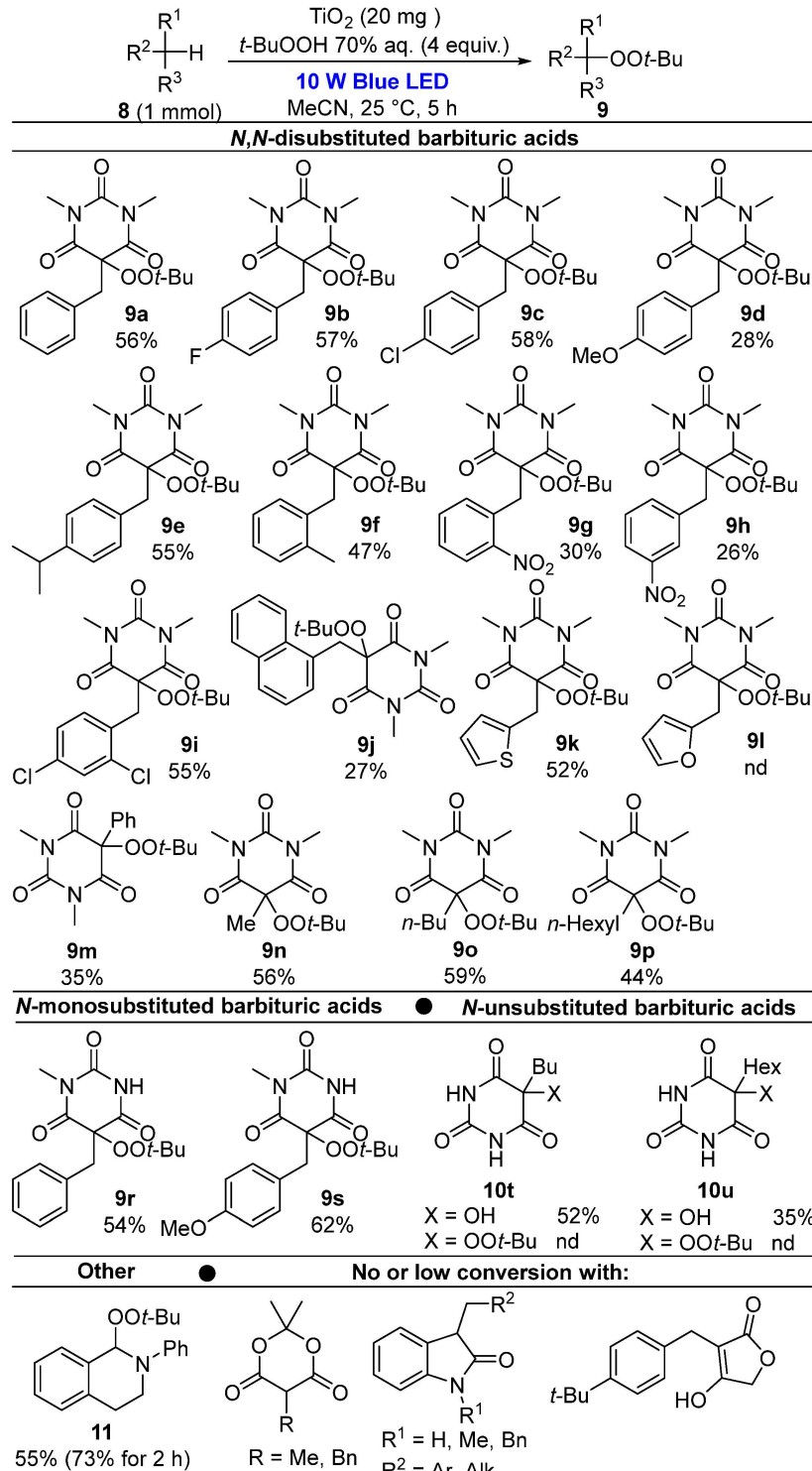

**Scheme 4.** Scope of substrates suitable for the peroxidation in a TBHP/TiO$_2$ system under visible light.

The peroxidation of N-substituted barbituric acids proceeds in moderate yields (27–62%), with the main by-product being the 5-hydroxy derivative. It should be noted separately that in the case of the substrate with the furanyl substituent **8l**, no product is formed, and the conversion of the initial substrate is insignificant. In the absence of substituents on both

nitrogen atoms, the target reaction is not observed; in this case, the main product is the 5-hydroxy product (**10t**, **10u**). In the next step, we tested some other classes of CH-reagents in peroxidation using the developed TBHP/TiO$_2$ photocatalytic system. Selective formation of the peroxide **11** from the tetrahydroisoquinoline is quite unusual, because TBHP is frequently used as an oxidant in reactions of tetrahydroisoquinolines with different nucleophiles [85,86], and the peroxide intermediate is easily intercepted by the nucleophile present in the system or oxidized to isoquinolin-1(2*H*)-one [87,88]. Other CH-acidic substrates, such as Meldrum's acids, oxindoles and cyclic β-ketoethers, demonstrate high stability to oxidation under reaction conditions, showing that our method is tolerant to easily oxidizable CH-acidic or enol groups.

To illustrate the substrate selectivity of the TBHP/TiO$_2$ peroxidation system, we conducted a series of experiments with competitive CH-acidic substrates (β-ketoester **12**, β-diketone **13** and malononitrile **14**, Scheme 5). In all cases, we obtained the desired peroxidation product **9a** with good yields (44–70%). With β-diketone **13**, we observed the incomplete conversion of barbituric acid **8a**, which led to a decrease in the **9a** yield.

**Scheme 5.** Experiments with competitive CH-acidic substrates.

Radical trapping EPR experiments employing DMPO (5,5-Dimethyl-1-pyrroline N-oxide) as a radical acceptor were conducted to study the formation of t-BuOOH-derived radicals in the *t*-BuOOH-TiO$_2$ system (see SI). It interesting to note that the radical formation in this system was observed even under dark conditions. However, the peroxidation reaction did not proceed without blue LED irradiation (Table 1, entry 2) and an increase in radical production was observed with EPR upon blue LED irradiation of the *t*-BuOOH-TiO$_2$ mixture in MeCN. The simulation of the observed DMPO radical adduct EPR spectrum employing the EasySpin 5.2.35 program [89] resulted in the following hyperfine splitting values of the main observed signal: $a_N = 1.31$ mT, $a_H^\beta = 1.04$ mT and $a_H^\gamma = 0.13$ mT. The obtained values are in agreement with those reported for the *t*-BuO• radical adduct to DMPO [90,91]; however, unambiguous assignment of the signal is not possible because very close values can be expected for *t*-BuO• and *t*-BuOO• adducts [92] (see the SI for details). Based on the experimental data and the existing literature, we proposed the following mechanism (Scheme 6).

**Scheme 6.** Plausible mechanism.

*Tert*-butylhydroperoxide decomposes on $TiO_2$ under visible light irradiation either by single-electron oxidation, leading to a *tert*-butylperoxyl radical (*t*-BuOO•), or single-electron reduction, leading to a *tert*-butoxyl radical (*t*-BuO•). The short-lived *tert*-butoxyl radical transforms to *t*-BuOH via hydrogen atom abstraction from TBHP with the additional formation of a *tert*-butylperoxyl radical. Thus, *t*-BuOO• radicals are main oxygen-centered radical species in the reaction mixture. Reactive *tert*-butylperoxyl and *tert*-butoxyl radicals abstract a H atom from barbituric acid **8a** with the formation of C-centered radical **15**, which is intercepted by *t*-BuOO• to give the desired product **9a**, stable under reaction conditions (see Table 2, entry 7). The C-centered radical **15** can be intercepted by molecular oxygen (derived from air or *t*-BuOOH decomposition) with the formation of hydroperoxyl radical **16**, which undergoes hydrogen atom transfer (HAT) from the medium, forming unstable hydroperoxide **17**. Hydroperoxide **17** decomposition leads to 5-hydroxy by-product **10a**.

## 3. Materials and Methods

### 3.1. General

Room temperature (rt) stands for 23–25 °C. $^{1}$H and $^{13}$C NMR spectra were recorded on Bruker (Bruker AXS Handheld Inc., Kennewick, WA, USA) AVANCE II 300 and Bruker Fourier 300HD (300.13 and 75.47 MHz, respectively) spectrometers in $CDCl_3$ and DMSO-D$_6$. Residual signals of $CDCl_3$ (7.26 in $^{1}$H NMR, 77.16 in $^{13}$C NMR) were used as reference signals for precise chemical shift determination. FT-IR spectra were recorded on Bruker Alpha instrument. IR spectra were registered in KBr pellets for solid compounds, and liquid compounds were placed between two KBr windows to make a thin layer. High-resolution mass spectra (HR-MS) were measured on a Bruker maXis instrument using electrospray ionization (ESI). The measurements were performed in a positive ion mode (interface capillary voltage—4500 V); mass range from $m/z$ 50 to $m/z$ 3000 Da; external calibration with Electrospray Calibrant Solution (Fluka). A syringe injection was used for all acetonitrile solutions (flow rate 3 µL/min). Nitrogen was applied as a dry gas; interface temperature was set at 180 °C.

Commercial $TiO_2$ Hombikat UV 100 (anatase, specific surface area, BET: 300 m$^2$·g$^{-1}$, primary crystal size according to Scherrer < 10 nm) was used as is. *Tert*-butyl peroxide (TBHP, 70% aqueous), cumyl hydroperoxide (CHP, 80%), dibenzoyl peroxide (75%), *meta*-chloroperoxybenzoic acid (mCPBA, 75%), di-*tert*-butylperoxide (DTBP, 99%) and 1,1,2,2-tetrachloroethane (98.5%) were used as is from commercial sources. Barbituric acids **1a–1x** were synthesized according to the literature procedures. Bulk g-$C_3N_4$ was prepared analogously to previously reported methods [93,94]: the urea was heated in a covered

alumina crucible for 4 h at 550 °C (heating rate 5 °C·min$^{-1}$). Dimethyl sulfoxide (DMSO), dimethyl formamide (DMF), acetic acid (AcOH) and 1,2-dichloroethane (DCE) were used as is from commercial sources. MeCN was distilled over $P_2O_5$. The reaction mixtures were sonicated in an ultrasonic bath (HF-Frequency 35 kHz, ultrasonic nominal power 80 W) before the irradiation.

### 3.2. Experimental Procedures

**Experimental details for Table 1**

Peroxide (1 mmol), $TiO_2$ (10 mg) and a solvent (MeCN, 2 mL) were placed in a 50 mL round-bottomed flask. The reaction mixture was sonicated for 2 min in an ultrasonic bath, then magnetically stirred in a thermostated water bath at 25 °C ($\pm 1$ °C) under irradiation of 10 W blue (443 nm) LED for 3 h. Then, the $C_2H_2Cl_4$ internal standard (40–60 mg) was added, and the reaction the mixture filtered through Celite. Yields and conversions were determined with $^1$H from the MeCN solutions using $C_2H_2Cl_4$ as the internal standard.

**Experimental details for Scheme 3**

1,1-diphehylethylene **3** (1 mmol, 180 mg) or 2,6-di-*tert*-butyl-4-methylphenol **6** (BHT, 1 mmol, 220 mg), $TiO_2$ (20 mg), *t*-BuOOH 70% aq. (4 mmol, 515 mg) and a solvent (MeCN, 2 mL) were placed in a round-bottomed flask. The reaction mixture was vacuumed and flushed with argon 3 times. The reaction mixture was sonicated for 2 min in an ultrasonic bath, then magnetically stirred in a thermostated water bath at 25 °C ($\pm 1$ °C) under irradiation of 10 W blue LED for 8 h. After reaction, the mixture was diluted with $CH_2Cl_2$ (10 mL) and poured into the water (20 mL). The layers were shaken, and the organic layer was separated. Water layer was extracted with $CH_2Cl_2$ (2 × 10 mL). Combined organic extracts were washed with brine (20 mL), dried over $MgSO_4$ and rotary-evaporated. Products **4** and **7** were isolated via column chromatography using Petroleum ether/EtOAc as eluent.

**Experimental details for Table 2**

Barbituric acid **8a** (0.5 mmol, 124 mg), $TiO_2$ (0–10 mg), *t*-BuOOH 70% aq. (0.5–2 mmol, 64–257 mg) and a solvent (1 mL) were placed in a round-bottomed flask. The reaction mixture was vacuumed and flushed with argon 3 times. The reaction mixture was sonicated for 2 min in an ultrasonic bath, then magnetically stirred in a thermostated water bath at 25 °C ($\pm 1$ °C) under irradiation of 10 W blue LED for 1–8 h. Then, the mixture was diluted with $CH_2Cl_2$ (10 mL) and poured into the water (20 mL). The layers were shaken, and the organic layer was separated. Water layer was extracted with $CH_2Cl_2$ (2 × 10 mL). Combined organic extracts were washed with brine (20 mL), dried over $MgSO_4$ and rotary-evaporated. Product **9a** was isolated via column chromatography using Petroleum ether/EtOAc = 1/5 as eluent.

**Experimental details for Scheme 4**

Barbituric acid **8** (1 mmol), $TiO_2$ (20 mg), *t*-BuOOH 70% aq. (4 mmol, 515 mg) and a solvent (MeCN, 2 mL) were placed in a round-bottomed flask. The reaction mixture was vacuumed and flushed with argon 3 times. The reaction mixture was sonicated for 2 min in an ultrasonic bath, then magnetically stirred in a thermostated water bath at 25 °C ($\pm 1$ °C) under irradiation of 10 W blue LED for 5 h. After reaction, the mixture was diluted with $CH_2Cl_2$ (10 mL) and poured into the water (20 mL). The layers were shaken, and the organic layer was separated. Water layer was extracted with $CH_2Cl_2$ (2 × 10 mL). Combined organic extracts were washed with brine (20 mL), dried over $MgSO_4$ and rotary-evaporated. Products **9a–10u** were isolated via column chromatography using Petroleum ether/EtOAc as eluent.

**Experimental details for Scheme 5**

Barbituric acid **8a** (0.5 mmol, 124 mg), competitive CH-substrate (ethyl 2-methyl-3-oxobutanoate, **12**, 0.5 mmol, 72 mg or 3-benzylpentane-2,4-dione, **13**, 0.5 mmol, 95 mg or 2-benzylmalononitrile **14**, 0.5 mmol, 78 mg), $TiO_2$ (10 mg), *t*-BuOOH 70% aq. (2 mmol, 257 mg) and MeCN (2 mL) were placed in a round-bottomed flask. The reaction mixture was vacuumed and flushed with argon 3 times. The reaction mixture was sonicated for

2 min in an ultrasonic bath, then magnetically stirred in a thermostated water bath at 25 °C ($\pm 1$ °C) under irradiation of 10 W blue LED for 5 h. Then, the reaction the mixture was diluted with $CH_2Cl_2$ (10 mL) and poured into the water (20 mL). The layers were shaken, and the organic layer was separated. Water layer was extracted with $CH_2Cl_2$ (2 × 10 mL). Combined organic extracts were washed with brine (20 mL), dried over $MgSO_4$ and rotary-evaporated. Product **9a** was isolated via column chromatography using Petroleum ether/EtOAc = 1/5 as eluent.

### 3.3. Characterization Data of the Synthesized Products

**(1,2-Bis(*tert*-butylperoxy)ethane-1,1-diyl)dibenzene 4** [95] was isolated via column chromatography (EtOAc/petroleum ether = 1/40, $R_f$ = 0.4) as a colorless liquid (30 mg, 8%). [1]H NMR (300 MHz, Chloroform-d) δ 7.43–7.37 (m, 4H), 7.36–7.27 (m, 6H), 4.84 (s, 2H), 1.24 (s, 9H), 1.12 (s, 9H). [13]C{[1]H}NMR (75.48 MHz, $CDCl_3$) δ 142.2, 127.9, 127.7, 127.5, 85.8, 80.6, 79.8, 77.8, 26.8, 26.4.

**2,6-Di-*tert*-butyl-4-(*tert*-butylperoxy)-4-methylcyclohexa-2,5-dien-1-one 7** was isolated via column chromatography (EtOAc/petroleum ether = 1/50, $R_f$ = 0.9) as yellow powder (43 mg, 14%). Mp = 84–85 °C (lit. Mp = 88–90 °C [53]). [1]H NMR (300 MHz, Chloroform-d) δ 6.55 (s, 2H), 1.31 (s, 3H), 1.21 (s, 18H), 1.17 (s, 9H). [13]C{[1]H}NMR (75.48 MHz, $CDCl_3$) δ 186.8, 146.8, 142.0, 79.5, 77.6, 76.3, 34.8, 29.6, 26.6, 24.4. HR-MS (ESI): $m/z$ = 331.2245, calcd. for $C_{19}H_{32}O_3$ + $Na^+$: 331.2244.

**5-Benzyl-5-(*tert*-butylperoxy)-1,3-dimethylpyrimidine-2,4,6(1H,3H,5H)-trione 9a** was isolated via column chromatography (EtOAc/petroleum ether = 1/5, $R_f$ = 0.6) as a white powder (187 mg, 56%). Mp = 93.0−94.0 °C (lit. Mp = 95 °C [53]). [1]H NMR (300 MHz, Chloroform-d) δ 7.25–7.18 (m, 3H), 7.03–6.95 (m, 2H), 3.26 (s, 2H), 3.10 (s, 6H), 1.23 (s, 9H). [13]C{[1]H}NMR (75.48 MHz, $CDCl_3$) δ 168.3, 150.2, 130.9, 129.6, 128.9, 128.4, 84.1, 82.4, 41.9, 28.6, 26.5. HR-MS (ESI): $m/z$ = 357.1416, calcd. for $C_{17}H_{22}N_2O_5$ + $Na^+$: 357.1421.

**5-(*Tert*-butylperoxy)-5-(4-fluorobenzyl)-1,3-dimethylpyrimidine-2,4,6(1H,3H,5H)-trione 9b** was isolated via column chromatography (EtOAc/petroleum ether = 1/5, $R_f$ = 0.5) as a white powder (200 mg, 57%). Mp = 102−103 °C (lit. Mp = 102−105 °C [53]). [1]H NMR (300 MHz, Chloroform-d) δ 7.07–6.84 (m, 4H), 3.25 (s, 2H), 3.13 (s, 6H), 1.21 (s, 9H). [13]C{[1]H}NMR (75.48 MHz, $CDCl_3$) δ 168.2, 162.7 (d, $J$ = 247.9 Hz), 150.2, 131.4 (d, $J$ = 8.2 Hz), 126.8 (d, $J$ = 3.6 Hz), 115.9 (d, $J$ = 21.4 Hz), 83.8, 82.4, 40.7, 28.6, 26.5. HR-MS (ESI): $m/z$ = 370.1771, calcd. for $C_{17}H_{21}FN_2O_5$ + $NH_4^+$: 370.1773.

**5-(*Tert*-butylperoxy)-5-(4-chlorobenzyl)-1,3-dimethylpyrimidine-2,4,6(1H,3H,5H)-trione 9c** was isolated via column chromatography (EtOAc/petroleum ether = 1/6, $R_f$ = 0.5) as a white powder (215 mg, 58%). Mp = 115–116 °C. [1]H NMR (300 MHz, Chloroform-d) δ 7.23–7.15 (m, 2H), 7.00–6.92 (m, 2H), 3.26 (s, 2H), 3.15 (s, 6H), 1.21 (s, 9H). [13]C{[1]H}NMR (75.48 MHz, $CDCl_3$) δ 168.1, 150.2, 134.5, 131.2, 129.6, 129.1, 83.6, 82.5, 40.7, 28.7, 26.5. FTIR (KBr): $\nu_{max}$ = 2980, 2937, 1705, 1686, 1491, 1445, 1429, 1380, 1367, 1305, 1294, 1231, 1198, 1178, 1116, 1092, 1068, 1031, 1016, 870, 845, 805, 752, 538, 501, 409 cm$^{-1}$. HR-MS (ESI): $m/z$ = 391.1035, calcd. for $C_{17}H_{21}ClN_2O_5$ + $Na^+$: 391.1031.

**5-(*Tert*-butylperoxy)-5-(4-methoxybenzyl)-1,3-dimethylpyrimidine-2,4,6(1H,3H,5H)-trione 9d** was isolated via column chromatography (EtOAc/petroleum ether = 1/10, $R_f$ = 0.3) as a white powder (103 mg, 28%). Mp = 120−121 °C (lit. Mp = 122−124 °C) [53]. [1]H NMR (300 MHz, Chloroform-d) δ 6.93–6.87 (m, 2H), 6.76–6.70 (m, 2H), 3.73 (s, 3H), 3.21 (s, 2H), 3.12 (s, 6H), 1.22 (s, 9H). [13]C{[1]H}NMR (75.48 MHz, $CDCl_3$) δ 168.4, 159.6, 150.3, 130.8, 122.6, 114.2, 84.1, 82.3, 55.3, 41.0, 28.6, 26.5. HR-MS (ESI): $m/z$ = 382.1974, calcd. for $C_{18}H_{24}N_2O_6$ + $NH_4^+$: 382.1973.

**5-(*Tert*-butylperoxy)-5-(4-isopropylbenzyl)-1,3-dimethylpyrimidine-2,4,6(1H,3H,5H)-trione 9e** was isolated via column chromatography (EtOAc/petroleum ether = 1/10, $R_f$ = 0.3) as a viscous liquid (206 mg, 55%) [53]. [1]H NMR (300 MHz, Chloroform-d) δ 7.09–7.00 (m, 2H), 6.93–6.82 (m, 2H), 3.20 (s, 2H), 3.07 (s, 6H), 2.79 (hept, $J$ = 6.9 Hz, 1H), 1.20 (s, 9H), 1.14 (d, $J$ = 6.9 Hz, 6H). [13]C{[1]H}NMR (75.48 MHz, $CDCl_3$) δ 168.3, 150.1, 149.1, 129.5, 128.0, 126.7, 84.2, 82.1, 41.6, 33.7, 28.4, 26.4, 23.9. HR-MS (ESI): $m/z$ = 394.2328, calcd. for $C_{20}H_{28}N_2O_5$ + $NH_4^+$: 394.2336.

**5-(*Tert*-butylperoxy)-1,3-dimethyl-5-(2-methylbenzyl)pyrimidine-2,4,6(1H,3H,5H)-trione 9f [53]** was isolated via column chromatography (EtOAc/petroleum ether = 1/10, $R_f$ = 0.4) as a colorless oil (162 mg, 47%). $^1$H NMR (300 MHz, Chloroform-*d*) δ 7.12–6.92 (m, 3H), 6.88–6.75 (m, 1H), 3.26 (s, 2H), 3.06 (s, 6H), 2.15 (s, 3H), 1.18 (s, 9H). $^{13}$C{$^1$H}NMR (75.48 MHz, CDCl$_3$) δ 168.2, 150.2, 137.5, 131.1, 130.1, 129.2, 128.3, 125.9, 83.9, 82.2, 38.6, 28.7, 26.5, 19.4. HR-MS (ESI): *m/z* = 366.2020, calcd. for C$_{18}$H$_{24}$N$_2$O$_5$ + NH$_4$$^+$: 366.2023.

**5-(*Tert*-butylperoxy)-1,3-dimethyl-5-(2-nitrobenzyl)pyrimidine-2,4,6(1H,3H,5H)-trione 9g** was isolated via column chromatography (EtOAc/petroleum ether = 1/5, $R_f$ = 0.3) as a white powder (116 mg, 30%). Mp = 139–141 °C (lit. Mp = 139–140 °C) [53]. $^1$H NMR (300 MHz, Chloroform-*d*) δ 7.98 (d, *J* = 8.1 Hz, 1H), 7.58 (t, *J* = 7.5 Hz, 1H), 7.51–7.38 (m, 2H), 3.66 (s, 2H), 3.28 (s, 6H), 1.17 (s, 9H). $^{13}$C{$^1$H}NMR (75.48 MHz, CDCl$_3$) δ 167.3, 150.7, 149.5, 134.4, 132.9, 129.2, 128.2, 125.2, 82.6, 81.7, 38.6, 29.1, 26.5. HR-MS (ESI): *m/z* = 397.1716, calcd. for C$_{17}$H$_{21}$N$_3$O$_7$ + NH$_4$$^+$: 397.1718.

**5-(*Tert*-butylperoxy)-1,3-dimethyl-5-(3-nitrobenzyl)pyrimidine-2,4,6(1H,3H,5H)-trione 9h** was isolated via column chromatography (EtOAc/petroleum ether = 1/5, $R_f$ = 0.7) as a colorless oil (100 mg, 26%). $^1$H NMR (300 MHz, Chloroform-*d*) δ 8.14–8.09 (m, 1H), 7.94–7.91 (m, 1H), 7.49–7.36 (m, 2H), 3.40 (s, 2H), 3.17 (s, 6H), 1.21 (s, 9H). $^{13}$C{$^1$H}NMR (75.48 MHz, CDCl$_3$) δ 167.7, 150.1, 148.4, 136.1, 133.5, 129.9, 124.7, 123.4, 83.0, 82.8, 40.6, 28.8, 26.4. HR-MS (ESI): *m/z* = 402.1273, calcd. for C$_{17}$H$_{21}$N$_3$O$_7$ + Na$^+$: 402.1272.

**5-(*Tert*-butylperoxy)-5-(2,4-dichlorobenzyl)-1,3-dimethylpyrimidine-2,4,6(1H,3H,5H)-trione 9i** was isolated via column chromatography (EtOAc/petroleum ether = 1/10, $R_f$ = 0.4) as a white powder (221 mg, 55%). Mp = 101–102 °C (lit. Mp = 101–103 °C) [53]. $^1$H NMR (300 MHz, Chloroform-*d*) δ 7.32 (d, *J* = 1.9 Hz, 1H), 7.16 (dd, *J* = 8.3, 2.0 Hz, 1H), 7.11 (d, *J* = 8.3 Hz, 1H), 3.37 (s, 2H), 3.21 (s, 7H), 1.21 (s, 9H). $^{13}$C{$^1$H}NMR (75.48 MHz, CDCl$_3$) δ 167.2, 150.5, 135.6, 134.9, 133.4, 129.6, 128.6, 127.2, 82.5, 38.6, 59.0, 26.5. HR-MS (ESI): *m/z* = 420.1077, calcd. for C$_{17}$H$_{20}$Cl$_2$N$_2$O$_5$ + NH$_4$$^+$: 420.1088.

**5-(*Tert*-butylperoxy)-1,3-dimethyl-5-(naphthalen-1-ylmethyl)pyrimidine-2,4,6(1H,3H,5H)-trione 9j** was isolated via column chromatography (EtOAc/petroleum ether = 1/5, $R_f$ = 0.6) as slightly yellow crystals (105 mg, 27%). Mp = 92–93 °C (lit. Mp = 92–93 °C) [53]. $^1$H NMR (300 MHz, Chloroform-*d*) δ 7.96–7.88 (m, 1H), 7.84–7.77 (m, 1H), 7.77–7.72 (m, 1H), 7.55–7.41 (m, 2H), 7.37–7.28 (m, 1H), 7.23–7.17 (m, 1H), 3.75 (s, 2H), 2.81 (s, 6H), 1.29 (s, 9H). $^{13}$C{$^1$H}NMR (75.48 MHz, CDCl$_3$) δ 168.2, 149.9, 133.8, 131.9, 129.3, 129.0, 128.7, 127.2, 126.5, 126.1, 124.9, 123.6, 84.2, 82.3, 38.5, 28.5, 26.6. HR-MS (ESI): *m/z* = 402.2016, calcd. for C$_{21}$H$_{24}$N$_2$O$_5$ + NH$_4$$^+$: 402.2023.

**5-(*Tert*-butylperoxy)-1,3-dimethyl-5-(thiophen-2-ylmethyl)pyrimidine-2,4,6(1H,3H,5H)-trione 9k** was isolated via column chromatography (EtOAc/petroleum ether = 1/5, $R_f$ = 0.7) as a white powder (177 mg, 52%). Mp = 84–85 °C (lit. Mp = 86–88 °C) [53]. $^1$H NMR (300 MHz, Chloroform-*d*) δ 7.19–7.11 (m, 1H), 6.91–6.84 (m, 1H), 6.75 (d, *J* = 3.5 Hz, 1H), 3.53 (s, 2H), 3.20 (s, 6H), 1.21 (s, 9H). $^{13}$C{$^1$H}NMR (75.48 MHz, CDCl$_3$) δ 168.2, 150.5, 132.0, 128.6, 127.4, 126.4, 83.7, 82.5, 35.5, 28.8, 26.5. HR-MS (ESI): *m/z* = 358.1429, calcd. for C$_{15}$H$_{20}$N$_2$O$_5$S + NH$_4$$^+$: 358.1431.

**5-(*Tert*-butylperoxy)-1,3-dimethyl-5-phenylpyrimidine-2,4,6(1H,3H,5H)-trione 9m** was isolated via column chromatography (EtOAc/petroleum ether = 1/4, $R_f$ = 0.5) as a white powder (113 mg, 35%). Mp = 112–113 °C. $^1$H NMR (300 MHz, Chloroform-*d*) δ 7.37 (m, 5H), 3.40 (s, 6H), 1.32 (s, 9H). $^{13}$C{$^1$H}NMR (75.48 MHz, CDCl$_3$) δ 167.7, 150.9, 130.4, 129.1, 126.7, 82.7, 29.3, 26.6. FTIR (KBr): $\nu_{max}$ = 2983, 1695, 1441, 1423, 1375, 1291, 1194, 1129, 1064, 1029, 867, 756, 717, 691, 634 cm$^{-1}$. HR-MS (ESI): *m/z* = 343.1271, calcd. for C$_{16}$H$_{20}$N$_2$O$_5$ + Na$^+$: 343.1264.

**5-(*Tert*-butylperoxy)-1,3,5-trimethylpyrimidine-2,4,6(1H,3H,5H)-trione 9n** was isolated via column chromatography (EtOAc/petroleum ether = 1/5, $R_f$ = 0.5) as a white powder (144 mg, 56%). Mp = 118–119 °C. $^1$H NMR (300 MHz, Chloroform-*d*) δ 3.34 (s, 6H), 1.60 (s, 3H), 1.18 (s, 9H). $^{13}$C{$^1$H}NMR (75.48 MHz, CDCl$_3$) δ 168.9, 151.0, 82.2, 79.3, 29.0, 26.4, 21.4. FTIR (KBr): $\nu_{max}$ = 2983, 1761, 1711, 1677, 1466, 1445, 1418, 1381, 1291, 1195, 1114, 1071, 870, 753 cm$^{-1}$. HR-MS (ESI): *m/z* = 281.1111, calcd. for C$_{11}$H$_{18}$N$_2$O$_5$ + Na$^+$: 281.1108.

**5-Butyl-5-(*tert*-butylperoxy)-1,3-dimethylpyrimidine-2,4,6(1H,3H,5H)-trione 9o [53]** was isolated via column chromatography (EtOAc/petroleum ether = 1/5, $R_f$ = 0.7) as a col-

orless liquid (162 mg, 59%). $^1$H NMR (300 MHz, Chloroform-*d*) δ 3.29 (s, 6H), 1.97–1.89 (m, 2H), 1.27–1.17 (m, 2H), 1.11 (s, 9H), 1.07–0.96 (m, 2H), 0.78 (t, *J* = 7.3 Hz, 3H). $^{13}$C{$^1$H}NMR (75.48 MHz, CDCl$_3$) δ 168.6, 150.9, 82.8, 81.9, 35.1, 28.7, 26.3, 24.9, 22.5, 13.6. HR-MS (ESI): *m*/*z* = 232.1578, calcd. for C$_{14}$H$_{24}$N$_2$O$_5$ + Na$^+$: 323.1577.

**5-(*Tert*-butylperoxy)-5-hexyl-1,3-dimethylpyrimidine-2,4,6(1H,3H,5H)-trione 9p** [51] was isolated via column chromatography (EtOAc/petroleum ether = 1/7, R$_f$ = 0.6) as a colorless liquid (152 mg, 44%). $^1$H NMR (300 MHz, Chloroform-*d*) δ 3.31 (s, 6H), 2.00–1.89 (m, 2H), 1.27–1.16 (m, 6H), 1.14 (s, 9H), 1.12–1.00 (m, 2H), 0.86–0.75 (m, 3H). $^{13}$C{$^1$H}NMR (75.48 MHz, CDCl$_3$) δ 168.7, 151.0, 82.9, 82.0, 35.4, 31.3, 29.0, 28.8, 26.4, 22.9, 22.5, 14.0. HR-MS (ESI): *m*/*z* = 346.2332, calcd. for C$_{16}$H$_{28}$N$_2$O$_5$ + NH$_4^+$: 346.2336.

**5-Benzyl-5-(*tert*-butylperoxy)-1-methylpyrimidine-2,4,6(1H,3H,5H)-trione 9r** was isolated via column chromatography (EtOAc/petroleum ether = 1/5, R$_f$ = 0.3) as a white powder (172 mg, 54%). Mp = 130–132 °C (lit. Mp = 132–133 °C, dec.) [53]. $^1$H NMR (300 MHz, Chloroform-*d*) δ 8.22 (s, 1H), 7.27–7.15 (m, 3H), 7.09–6.97 (m, 2H), 3.28 (s, 2H), 3.07 (s, 3H), 1.21 (s, 9H). $^{13}$C{$^1$H}NMR (75.48 MHz, CDCl$_3$) δ 169.0, 167.6, 149.0, 130.6, 130.0, 129.0, 128.5, 84.1, 82.6, 41.1, 28.0, 26.5. HR-MS (ESI): *m*/*z* = 343.1266, calcd. for C$_{16}$H$_{20}$N$_2$O$_5$ + Na$^+$: 343.1264.

**5-(*Tert*-butylperoxy)-5-(4-methoxybenzyl)-1-methylpyrimidine-2,4,6(1H,3H,5H)-trione 9s** was isolated via column chromatography (EtOAc/petroleum ether = 1/4, R$_f$ = 0.3) as a white powder (230 mg, 62%). Mp = 137–138 °C. $^1$H NMR (300 MHz, Chloroform-*d*) δ 8.92 (s, 1H), 6.95 (d, *J* = 8.6 Hz, 2H), 6.72 (d, *J* = 8.7 Hz, 2H), 3.70 (s, 3H), 3.22 (s, 2H), 3.08 (s, 3H), 1.20 (s, 9H). $^{13}$C{$^1$H}NMR (75.48 MHz, CDCl$_3$) δ 169.1, 168.0, 159.5, 149.3, 131.0, 122.2, 114.3, 84.0, 82.4, 55.2, 40.2, 27.9, 26.4. FTIR (KBr): ν$_{max}$ = 3473, 3415, 3244, 2980, 1763, 1724, 1697, 1613, 1514, 1447, 1386, 1368, 1305, 1253, 1183, 1073, 1031, 870, 844, 820, 793, 532 cm$^{-1}$. HR-MS (ESI): *m*/*z* = 373.1380, calcd. for C$_{17}$H$_{22}$N$_2$O$_6$ + Na$^+$: 373.1370.

**5-Benzyl-5-hydroxy-1,3-dimethylpyrimidine-2,4,6(1H,3H,5H)-trione 10a** was isolated via column chromatography (EtOAc/petroleum ether = 1/5, R$_f$ = 0.1) as a white powder. Mp = 112–114 °C (lit. Mp = 113–114 °C [96]). $^1$H NMR (300 MHz, Chloroform-*d*) δ 7.28–7.16 (m, 3H), 6.96–6.86 (m, 2H), 3.75 (bs, 1H), 3.21 (s, 2H), 3.06 (s, 6H). $^{13}$C{$^1$H}NMR (75.48 MHz, CDCl$_3$) δ 169.9, 149.9, 132.0, 129.3, 128.8, 128.6, 77.1, 50.0, 28.6. HR-MS (ESI): *m*/*z* = 285.0844, calcd. for C$_{13}$H$_{14}$N$_2$O$_4$ + Na$^+$: 285.0846.

**5-butyl-5-hydroxypyrimidine-2,4,6(1H,3H,5H)-trione 10t** was isolated via column chromatography (EtOAc/DCM = 1/1, R$_f$ = 0.4) as a white powder (141 mg, 52%). Mp = 175–177 °C. $^1$H NMR (300 MHz, DMSO-*d$_6$*) δ 11.26 (s, 2H), 6.00 (s, 1H), 1.88–1.64 (m, 1H), 1.40–1.08 (m, 2H), 0.81 (t, *J* = 6.8 Hz, 1H).$^{13}$C{$^1$H}NMR (75.48 MHz, CDCl$_3$) δ 172.2, 150.0, 74.8, 39.0, 24.8, 22.0, 13.8. FTIR (KBr): ν$_{max}$ = 3396, 3301, 3219, 3101, 2961, 2873, 1759, 1708, 1436, 1419, 1405, 1375, 1320, 1266, 1246, 1221, 1167, 1124, 1098, 822, 762, 528, 500 cm$^{-1}$. HR-MS (ESI): *m*/*z* = 223.0688, calcd. for C$_8$H$_{12}$N$_2$O$_5$ + Na$^+$: 223.0689.

**5-Hexyl-5-hydroxypyrimidine-2,4,6(1H,3H,5H)-trione 10u** was isolated via column chromatography (EtOAc/DCM = 1/1, R$_f$ = 0.5) as a white powder (84 mg, 35%). Mp = 128–129 °C. $^1$H NMR (300 MHz, DMSO-*d$_6$*) δ 11.24 (s, 2H), 5.98 (s, 1H), 1.81–1.72 (m, 2H), 1.28–1.14 (m, 8H), 0.83 (t, *J* = 6.3 Hz, 3H). $^{13}$C{$^1$H}NMR (75.48 MHz, CDCl$_3$) δ 172.0, 149.9, 74.7, 39.2, 30.9, 28.4, 22.5, 21.9, 13.8. FTIR (KBr): ν$_{max}$ = 3483, 3219, 3073, 2959, 2922, 2855, 1740, 1698, 1436, 1368, 1316, 1239, 1223, 1208, 1164, 1094, 796, 500 cm$^{-1}$. HR-MS (ESI): *m*/*z* = 251.1003, calcd. for C$_{10}$H$_{14}$N$_2$O$_5$ + Na$^+$: 251.1002.

**1-(*Tert*-butylperoxy)-2-phenyl-1,2,3,4-tetrahydroisoquinoline 11** [97] was isolated via column chromatography (EtOAc/petroleum ether = 1/7, R$_f$ = 0.8) as slightly yellow crystals (163 mg, 55%). Mp = 60–61 °C. $^1$H NMR (300 MHz, Chloroform-*d*) δ 7.42–7.36 (m, 1H), 7.34–7.18 (m, 5H), 7.18–7.13 (m, 2H), 6.90–6.82 (m, 1H), 6.21 (s, 1H), 3.81–3.67 (m, 1H), 3.64–3.50 (m, 1H), 3.17–2.90 (m, 2H), 1.15 (s, 9H). $^{13}$C{$^1$H}NMR (75.48 MHz, CDCl$_3$) δ 149.0, 136.7, 133.1, 129.2, 129.1, 128.7, 126.1, 119.0, 115.0, 90.8, 80.1, 42.7, 28.3, 26.7. HR-MS (ESI): *m*/*z* = 331.2245, calcd. for C$_{19}$H$_{32}$O$_3$ + Na$^+$: 331.2244.

## 4. Conclusions

It was found that commercial unmodified $TiO_2$ can catalyze the decomposition of organic hydroperoxides (e.g., *tert*-butylhydroperoxide, cumyl hydroperoxide) under visible light irradiation (443 nm). The *t*-BuOOH/$TiO_2$ photocatalytic system can be used for the generation of *tert*-butylperoxyl radicals under mild conditions. The synthetic application of the *t*-BuOOH/$TiO_2$ system was demonstrated by the peroxidation of barbituric acids. The peroxidation is highly chemoselective and can proceed in the presence of weak benzylic CH-bonds or CH-acidic substrates such as Meldrum's acids, β-ketoesters, β-diketones and malononitriles.

**Supplementary Materials:** The following supporting information can be downloaded at: https://www.mdpi.com/article/10.3390/catal13091306/s1, Figure S1: EPR spectra of spin trapping experiments under air. Black lines—EPR spectra before irradiation, Figure S2: EPR spectra of spin trapping experiments under argon atmosphere, Figure S3: Experimental (black) and simulated (red) EPR spectra for mixtures in MeCN, Table S1: Simulation parameters for spectra presented in Figure S3 in comparison with literature data, Figure S4: Appearance of $TiO_2$ suspension in MeCN in the presence (left) and absence (right) of *t*-BuOOH, Figure S5: UV-Vis spectrum of 10W Blue LED used for photochemical syntheses in the present study (λmax = 443 nm), Spectral data of the synthesized compounds.

**Author Contributions:** Conceptualization, I.B.K.; methodology, I.B.K. and E.R.L.; investigation, E.R.L.; writing—original draft preparation, E.R.L.; writing—review and editing, I.B.K. and A.O.T.; supervision, I.B.K. and A.O.T.; project administration, A.O.T. All authors have read and agreed to the published version of the manuscript.

**Funding:** This work was supported by the Russian Science Foundation (Grant No. 21-43-04417).

**Data Availability Statement:** Not applicable.

**Conflicts of Interest:** The authors declare no conflict of interest.

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
