# Peer review of "t-BuOOH/TiO2 Photocatalytic System as a Convenient Peroxyl Radical Source at Room Temperature under Visible Light and Its Application for the CH-Peroxidation of Barbituric Acids"

_catalysts, doi:10.3390/catal13091306_

Round 1

Reviewer 1 Report

In the paper entitled "t-BuOOH / TiO2 photocatalytic system as a convenient peroxyl radical source at room temperature under visible light and its application for CH-peroxidation of Barbituric acids" Lopat’eva and co-workers described the heterogeneous photocatalysis with the use of titania irradiated at 443 nm (blue light) in the presence of oxidizing factor for barbituric acid peroxidation. The proposed synthetic method yield over 20 new peroxidized derivatives. Although the presented results seem attractive, the manuscript should be revised before publication:

1. The advance of the irradiation of blue light instead of UV light in the production of radicals on the surface of titania should be presented in the introduction.

2. Table 1 - why has H2O2 not been chosen for the photodecomposition of organic peroxides on TiO2 under blue light irradiation?

3. The formation of two types of radicals (tert-butylperoxyl and tert-butoxyl) should be deeply investigated using EPR to determine their ratio.

4. The characterization of some compounds in section 3.3 has to be completed with HR-MS analysis for all compounds, not only for selected ones (e.g. 9c, 9h, 9m etc.).

5. 13C NMR spectrum of compound 9b - Please zoom in on the aromatic region of the NMR spectrum to show the splitting of the carbon signal at the fluorine substituent.

Author Response

(The step-by-step answers to all comments are also attached as PDF file) 

Reviewer’s comment: In the paper entitled "t-BuOOH / TiO2 photocatalytic system as a convenient peroxyl radical source at room temperature under visible light and its application for CH-peroxidation of Barbituric acids" Lopat’eva and co-workers described the heterogeneous photocatalysis with the use of titania irradiated at 443 nm (blue light) in the presence of oxidizing factor for barbituric acid peroxidation. The proposed synthetic method yield over 20 new peroxidized derivatives. Although the presented results seem attractive, the manuscript should be revised before publication:

Answer: Thank you for careful reading and high evaluation of our work

Reviewer’s comment: 1. The advance of the irradiation of blue light instead of UV light in the production of radicals on the surface of titania should be presented in the introduction.

Answer: Thank you for the valuable advice. The following text was added to the introduction:

“Visible light is an attractive energy source for chemical transformations as it is responsible for the greatest part of the sunlight irradiation power compared to UV light. UV-mediated photochemistry is frequently associated with additional safety precautions, expensive light sources (compared to visible light), and the need for UV-transparent quartz glassware. Moreover, most organic compounds, including solvents, absorb UV radiation, which leads to lower energy efficiency and unwanted photochemical side reactions. Visible light quanta have lower energy and thus more promising for the development of selective and widely applicable photochemical synthetic methods based on suitable photocatalysts.”

Reviewer’s comment: 2. Table 1 - why has H2O2 not been chosen for the photodecomposition of organic peroxides on TiO2 under blue light irradiation?

Answer: It is known that hydrogen peroxide can be decomposed under UV radiation on the TiO2 surface to form hydroxyl radicals [10.1016/j.apcatb.2015.12.044], which is widely used to accelerate the photodegradation of various pollutants [10.1016/j.jhazmat.2006.06.093, 10.1016/S1010-6030(02)00004-7, 10.1021/es020898n, 10.1016/j.cattod.2008.10.003, 10.1016/j.watres.2003.10.037, 10.1016/j.desal.2009.11.003]. On the other hand, modified visible-light active TiO2 can be used for H2O2 synthesis rather than oxidation of pollutants [10.3390/molecules26133844, 10.1016/j.cej.2010.11.093, 10.1039/D0CC03327H, 10.3390/catal9070623, 10.1016/j.catcom.2021.106315, etc.]. Based on these literature data, the decomposition of hydrogen peroxide on the TiO2 surface would require UV radiation. Introduction of H2O2 to the reaction mixture can result in additional side oxidation processes, thus, we used only one organic peroxide in each experiment.

Reviewer’s comment: 3. The formation of two types of radicals (tert-butylperoxyl and tert-butoxyl) should be deeply investigated using EPR to determine their ratio.

Answer: Thank you for this important proposal. According to the Reviewer’s advice, we have conducted extensive EPR study of t-BuOOH-TiO2 system, and these experiments confirmed the formation of O-centered radicals from t-BuOOH. However, tert-butylperoxyl and tert-butoxyl ratio can not be estimated (see supporting information for the details). Following text was added to the main text:

“Radical trapping EPR experiments employing DMPO (5,5-Dimethyl-1-pyrroline N-oxide) as radical acceptor were conducted to study the formation of t-BuOOH-derived radicals in t-BuOOH-TiO2 system (see SI). It interesting to note that the radical formation in this system was observed even under dark conditions, however, peroxidation reaction do not proceed without blue LED irradiation (Table 1, entry 2) and increase in radical production is observed by EPR upon blue LED irradiation of t-BuOOH-TiO2 mixture in MeCN. The simulation of observed DMPO radical adduct EPR spectrum employing EasySpin 5.2.35 program89 resulted in following hyperfine splitting values of main observed signal aN = 1.31 mT, aHβ = 1.04 mT, aHγ = 0.13 mT. Obtained values are in agreement with those reported for t-BuO• radical adduct to DMPO,90,91 however, unambiguous assignment of signal is not possible because very close values can be expected for t-BuO•  and t-BuOO• adducts92 (see SI for details).

Reviewer’s comment: 4. The characterization of some compounds in section 3.3 has to be completed with HR-MS analysis for all compounds, not only for selected ones (e.g. 9c, 9h, 9m etc.).

Answer: We added HR-MS for all the synthesized compounds. The peroxide 4 was reported previously, and all the required information (1H, 13C NMR, Rf) agrees with the literature data. Registration of mass spectra of this particular peroxide is difficult due to its decomposition under HR-MS conditions.

Reviewer’s comment: 5. 13C NMR spectrum of compound 9b - Please zoom in on the aromatic region of the NMR spectrum to show the splitting of the carbon signal at the fluorine substituent.

Answer: The region 116–132 ppm was expanded.

Reviewer 2 Report

In this study, t-BuOOH/TiO2 photocatalyst integrated with visible light served as a convenient system for generation of peroxyl radicals. This topic seems to be novel for application of commercial TiO2. However, some viewpoints need to be clarified.

1.     It is well known that TiO2 could be activated by UV light. However, in scheme 2, TiO2 was effectively activated by visible light for generation of free electrons. Please supply some references to support the hypothesis.

2.     In Table 2, it seems that TiO2 is effectively activated by visible light. Please provide the light emission pattern of 10 W Blue LED.

3.     In Table 2, optimization of amounts of t-BuOOH, solvent, and reaction time was carried out. Please indicate the optimal conditions in the text.

4.     The amounts of peroxyl radicals generated depend on t-BuOOH concentrations, TiO2 dosages, reaction time. Please provide testing data for higher concentrations and dosages of t-BuOOH and TiO2 in Table 2.

On the basis of above discussion, this manuscript is recommended to be minor revised. 

Author Response

(The step-by-step answers to all comments are also attached as PDF file) 

Reviewer’s comment: In this study, t-BuOOH/TiO2 photocatalyst integrated with visible light served as a convenient system for generation of peroxyl radicals. This topic seems to be novel for application of commercial TiO2. However, some viewpoints need to be clarified.

Answer:  Thank you for the high evaluation of our work

Reviewer’s comment: 1. It is well known that TiO2 could be activated by UV light. However, in scheme 2, TiO2 was effectively activated by visible light for generation of free electrons. Please supply some references to support the hypothesis.

Answer: Thank you for the insightful comment. Indeed, TiO2 is known to be activated by UV light. The following text was added to the introduction:

«Numerous attempts were directed to the photosensitization of titanium dioxide in the visible light region by another semiconductors,18–21 organic photocatalysts22 or by element-doping.23–28 Another opportunity is to use organic compounds, especially bearing hydroxyl or carboxyl groups, such as phenols,29,30 salicylic acid,31,32 carboxylic acids33–35 or N-hydroxyphthalimide,36,37 which can modify the TiO2 surface to absorb visible light. In our work we discovered that peroxides can also lead to visible light sensitization: the mixing of TiO2 suspension in MeCN with tert-butylhydroperoxide resulted in change in color from white to pale yellow (The appearance of the mixture can be found in SI).»

A slight color change from white to pale yellow was observed when t-BuOOH 70% aq. was added to TiO2 suspension in MeCN – the image of the suspension was added to the SI (see Figure S4 on page S10).

Reviewer’s comment: 2. In Table 2, it seems that TiO2 is effectively activated by visible light. Please provide the light emission pattern of 10 W Blue LED.

Answer: The light emission spectrum of 10 W Blue LED was added to the SI (see Figure S5 on page S10).

Reviewer’s comment: 3. In Table 2, optimization of amounts of t-BuOOH, solvent, and reaction time was carried out. Please indicate the optimal conditions in the text.

Answer: The following text was added:

“It turns out that scaling up to 1 mmol of 8a led to the slight increase in 9a yield, therefore the conditions of entry 18 were chosen as optimal for the peroxidation of other barbituric acids.”

Reviewer’s comment: 4. The amounts of peroxyl radicals generated depend on t-BuOOH concentrations, TiO2 dosages, reaction time. Please provide testing data for higher concentrations and dosages of t-BuOOH and TiO2 in Table 2.

Answer: The increase in TiO2 dosage from 10 mg (standard conditions) to 20 mg (entry 11) led to slight decrease in 9a yield. Experiment with 6 equiv. of t-BuOOH was added to the Table 2 (Enty 10).

Round 2

Reviewer 1 Report

Dear Authors,

All my issues have been addressed. I have no further comments. The paper can be accepted in its present form.